# The 10 Year Outcomes of Implants Inserted with Dehiscence or Fenestrations in the Rehabilitation of Completely Edentulous Jaws with the All-on-4 Concept

**DOI:** 10.3390/jcm11071939

**Published:** 2022-03-31

**Authors:** Miguel de Araújo Nobre, Armando Lopes, Elsa Antunes

**Affiliations:** 1Research, Development and Education Department, Maló Clinic, 1600-042 Lisboa, Portugal; 2Oral Surgery Department, Maló Clinic, 1600-042 Lisboa, Portugal; alopes@maloclinics.com; 3Dental Hygiene Department, Maló Clinic, 1600-042 Lisboa, Portugal; eantunes@maloclinics.com

**Keywords:** dental implants, All-on-4, full-arch, immediate function, immediate loading, dehiscence, fenestrations

## Abstract

Background: There is a need for a long-term evidence of implants placed in challenging conditions. The aim of this study was to investigate the outcome of full-arch rehabilitations with the All-on-4 concept for implants inserted with dehiscence or fenestrations. Methods: This retrospective cohort study included 123 patients (dehiscence, *n* = 87 patients; fenestrations, *n* = 28 patients; both conditions, *n* = 8 patients), with a total of 192 implants in immediate function presenting dehiscence (*n* = 150), fenestrations (*n* = 40), or both conditions (*n* = 2). Primary outcome measures were cumulative implant survival (CSurR) and success (CSucR) rates. Secondary outcome measures were prosthetic survival, marginal bone loss, and incidence of biological complications. Results: CSurRs were 94.1% (overall), 95.6% (dehiscence), and 88.1% (fenestrations) at 10 years using the patient as the unit of analysis. Smoking affected implant failure significantly (*p* = 0.019). Implant-level CSurRs and CSucRs at 10 years were 96.2% and 93.5% (overall), 97.2% and 94.6% (dehiscence), and 90.0% and 87.6% (fenestrations), respectively. Average bone resorption at 5 and 10 years was 1.22 mm and 1.53 mm, respectively. Biological complications occurred in 18 patients (*n* = 18 implants). Conclusions: Implants inserted with dehiscence or fenestrations demonstrate good long-term outcomes with overall high success and survival rates and low average marginal bone resorption, despite an inferior outcome in implants with fenestrations and smoking’s negative effect.

## 1. Introduction

Immediate function dental implants used for the support of fixed prosthetic rehabilitations provide successful long-term outcomes considering the high implant survival rates and patient satisfaction [1,2,3,4,5]. The All-on-4 concept (Nobel Biocare, Gothenburg, Sweden) is a rehabilitation protocol for fixed prosthetic rehabilitation of edentulous arches supported by implants in immediate function (two anterior implants placed in an axial orientation and two posterior implants tilted distally) with documented high survival rates in the long term for both the maxilla (95.7% at 13 years) [6] and the mandible (91.7% at 18 years) [7].

Frequently, the inclusion criteria for rehabilitation imply the insertion of implants in conditions close to the ideal: sufficient bone quantity and quality, with absence of infection on the implant site. Nevertheless, in particular situations, implants are inserted in challenging conditions, including post-extraction sockets, on periodontally compromised sites, or in sites presenting low bone density and quantity, with the possibility of dehiscence or a fenestration occurring [8,9,10,11].

In implant dentistry, dehiscence is defined as the absence of alveolar bone on the buccal or palatine/lingual aspects, leaving an oval-shaped defect and implant exposure apically from the implant–abutment junction (Figure 1); in turn, a fenestration is defined as a buccal or lingual window defect leaving marginal bone in situ (Figure 2). When inserting the implant, the presence of dehiscence or a fenestration can negatively impact the long-term outcome by decreased bone support mucosal irritation [12] and increased marginal bone strain on the implant’s mesial and distal sites (based on finite element analysis) [13]. The published reports on the outcomes of implants placed in sites presenting dehiscence are inconclusive. High survival rates in terms of short- [14,15], mid- [16,17], and long-term [18] outcomes have been reported. Nevertheless, a number of complications with the potential to decrease success have also been reported, including increased attachment loss, deeper peri-implant pockets [15], or postoperative exposure/infection when using concomitant guided bone regeneration [18]. In severely resorbed arches, fenestrations have a higher probability of occurrence given the insufficient bone volume [8]. The existence of fenestrations may be successfully managed through guided bone regeneration to the point of achieving a similar short-term outcome compared to implants placed in healed sites [19] but there is still lack of robust evidence to determine if any treatment is needed and which is the best treatment [8,20,21].

Considering the literature, it is noticeable that there is a need for mid- and long-term studies evaluating the outcome of dental implants inserted with dehiscence and fenestrations. The aim of this study was to document the 5-year and 10-year outcomes of full-arch rehabilitations with the All-on-4 concept supported by implants in the presence of dehiscence and/or fenestrations at the moment of implant placement. The research hypothesis investigated in this study was that the outcome of immediate-function implants inserted with dehiscence or fenestrations does not differ from the outcome of immediate-function implants inserted in nearly ideal conditions (sufficient bone quantity and quality, and no infection at the implant site).

## 2. Materials and Methods

This clinical study was performed complying with all ethical regulations in accordance with the Declaration of Helsinki and approved by an Ethical Committee (Ethical Committee for Health, Lisbon, Portugal; authorization no. 002/2017).

A database of patients with full-arch rehabilitations with the All-on-4 concept (Nobel Biocare) between April 2003 and June 2010 was assessed. The patients were selected on the basis of full-arch restorations supported by dental implants in the immediate function of the edentulous maxilla and/or mandible; the possibility of placing a minimum of four implants (at least 8.5 mm long) into the completely edentulous arch using a tilted approach for the distal implants; and the presence of dehiscence or fenestrations confirmed clinically and perioperatively, complying with a convenience sample. Exclusion criteria included active maxillary radiation or chemotherapy; systemic condition or smoking habits were not considered as exclusion criteria.

The surgical protocol followed the All-on-4 concept configuration (Nobel Biocare AB), refs. [6,7] with the two posterior implants inserted with distal tilting and the two anterior implants inserted axially. The implant insertion followed an under-preparation drilling protocol to guarantee a minimum of 35 N·cm insertion torque. The preparation was typically performed to the full drill depth with a 2 mm twist drill followed by a 2.4/2.8 mm step drill and a 3.2/3.6 mm step drill (depending on bone density). In sites with dense bone, 3.8/4.2 mm step drills were used. The implant neck was positioned at the bone level, and bicortical anchorage was established whenever possible in the maxilla.

Dehiscence occurred in certain sites with reduced crest thickness during implant preparation and/or placement, mainly on the lingual/palatal aspect of the crest because the implant site preparations were made more lingually than buccally to ensure thicker buccal cortical bone for protection and blood supply. In these situations, no particular protocol was used to minimize or restore dehiscence, in which, after implant placement, the flap was sutured back against the bone. Considering fenestrations, these were mainly accidental in cases where the buccal bone anatomy was irregular with reduced crest thickness (e.g., concavities) and in immediate extraction cases when the tilted implants came into contact with the fresh sockets. In the first situation, no special care was taken, with the flap sutured back against the bone; in the second situation, if the implant was exposed inside a socket, an autogenous bone graft, obtained during smoothing of the bone crest, was condensed in the socket to hide the fenestration and prevent soft-tissue migration.

Concerning the prosthetic protocol, a fixed high-density provisional prosthesis composed of acrylic resin (PalaXpress Ultra; Heraeus Kulzer GmbH, Hanau, Germany) with titanium cylinders (Nobel Biocare AB) and acrylic resin prosthetic teeth (Mondial and Premium teeth, Heraeus Kulzer GmbH) was manufactured at the dental laboratory and connected on the day of surgery.

Considering patient desires, a metal ceramic implant-supported fixed prosthesis with a titanium framework and all-ceramic crowns (NobelProcera titanium framework, NobelProcera crowns, Nobel Rondo ceramics; Nobel Biocare AB) or a metal–acrylic resin implant-supported fixed prosthesis with a titanium framework (NobelProcera titanium framework; Nobel Biocare AB) and acrylic resin prosthetic teeth (Mondial and Premium teeth, Heraeus Kulzer GmbH) were used to replace the provisional prosthesis. The final prosthesis was delivered typically 6 months post-surgically. A clinical situation is illustrated in Figure 3.

A postoperative maintenance protocol was indicated for each patient, including oral hygiene instructions. Follow-up clinical appointments were performed at 10 days, 2, 4, and 6 months, 1 year, and every 6 months thereafter, consisting of the assessment of clinical parameters, prophylaxis, and dental hygiene instructions. The evaluators were calibrated, with intra- and inter-class correlation coefficients of 0.92 and 0.85, respectively (weighted Kappa scores).

### 2.1. Outcome Measures

An outcome assessor blinded to the objectives of the study evaluated the data. Outcomes were assessed at implant surgery, as well as at 5 and 10 years of follow-up. The primary outcome measure was implant success, according to the success criteria adopted by the authors [22]: (a) The implant fulfilled its intended function supporting the reconstruction (sleeping implants were considered failures); (b) implant was stable upon manual testing; (c) absence of persistent infection jeopardizing the implant outcome; (d) no areas of radiolucency around the implants; (e) good esthetic outcome; (f) construction of an implant-supported fixed restoration that was comfortable for the patient and with conditions for good hygienic maintenance. Implants not complying with the criteria were considered survivals. Implant removal was classified as failure.

Secondary outcome measures were prosthetic survival (based on function with the necessity of replacing the prosthesis classified as failure), marginal bone loss at 5 and 10 years, and the incidence of biological complications.

The radiographic evaluation used to assess marginal bone loss was performed at baseline, as well as at 5 and 10 years of follow-up, using periapical radiographs by utilizing the parallelometric intraoral technique. For the intraoral technique, a conventional radiograph holder was used, the position of which was adjusted manually to ensure orthogonal film positioning. A blinded operator examined all radiographs of the implants for the marginal bone level. Each periapical radiograph was scanned at 300 dpi with a scanner (HP Scanjet 4890, HP Portugal, Paço de Arcos, Portugal). The marginal bone level was assessed with image analysis software (Image J version 1.40 g for Windows, National Institutes of Health, Bethesda, MD, USA) using the implants’ inter-thread distance as the reference for digital calibration. The implant platform was used as a reference point, and marginal bone loss was defined as the difference in marginal bone levels between the day of surgery and the point of evaluation. The radiographs were accepted or rejected for evaluation on the basis of the clarity of implant threads; a clear thread guarantees both sharpness and an orthogonal direction of the radiographic beam toward the implant axis.

The evaluation of biological complications included peri-implant pathology (defined as peri-implant pocket depths ≥5 mm, bleeding on probing, with concurrent marginal bone loss compared to the previous radiograph or clinical attachment loss of >2 mm) [23], fistula formation, or abscess. The evaluation was performed using a plastic periodontal probe calibrated to 0.25 N (Click Probe, Hawe Neos, Bioggio, Switzerland).

The overall medical status was assessed from patient records by interviewing the patients, and the distribution was classified according to the International Classification of Diseases 11th Revision (ICD-11) [24]. Bruxism, in particular, was assessed as per protocol by interviewing the patient (about symptoms related to tenderness of jaw muscles, sleeping habits, and medication), observation of natural teeth (if present) or prosthetic elements (if present), and radiographic evaluation.

### 2.2. Statistical Evaluation

Descriptive statistics (average, standard deviation, and range) were calculated for age and marginal bone loss (at 5 and 10 years). Frequencies were used to classify biological complications, loss to follow-up, and prosthetic survival. Cumulative implant survival and success were estimated at the patient level (any implant failure in each patient) by using the Kaplan–Meier product limit estimator and at the implant level using life tables. Inferential analysis was performed to evaluate the difference in demographics between patients with complete follow-up and patients lost to follow-up (age: Mann–Whitney U test; sex: chi-square test). Data were statistically analyzed using the Statistical Package for the Social Sciences software (IBM SPSS, version 17, Rochester, NY, USA).

## 3. Results

### 3.1. Sample

A total of 123 patients (38 males and 85 females; mean age, 55.2 years; range, 34–81 years) with 127 full-arch rehabilitations (maxilla: 84; mandible: 43) were included (dehiscence, *n* = 87 patients; fenestrations, *n* = 28 patients; dehiscence and fenestrations, *n* = 8 patients). Considering the International Classification of Diseases 11th Revision (ICD-11) [24], 79 patients had health complications (Table 1). A total of 192 implants were inserted (Table 1 and Table 2): 150 implants with dehiscence (maxilla: 100 implants; mandible: 50 implants), 40 implants with a fenestration (maxilla: 30 implants; mandible: 10 implants), and 2 implants with dehiscence and a fenestration (maxilla: 1 implant; mandible: 1 implant). The opposing dentitions were implant-supported prostheses (57 patients), natural teeth (19 patients), a combination of both (37 patients), fixed prosthesis over natural teeth (1 patient), and removable prosthesis (9 patients).

### 3.2. Lost to Follow-Up Rate; Implant Survival, Success, and Failure; Prosthetic Survival

Twenty-five patients (20%) with 34 implants (17.8%) became unreachable and were lost to follow-up (dehiscence: 19 patients and 27 implants; fenestrations: 5 patients with 5 implants; dehiscence and fenestrations: 1 patient and 2 implants). Comparing demographic variables, a significant difference was registered for age with an increased age for patients lost to follow-up compared to patients with complete follow-up (*p* = 0.014, Mann–Whitney U test), while no significant difference was registered for sex (*p* = 0.647, chi-square test). Seven patients lost seven implants (one implant in each patient; dehiscence: three implants; fenestrations: three implants; dehiscence and fenestration: one implant; Table 1 and Table 2). In three patients, new implants were inserted (not accounted for in this study) to replace the failed implants; two patients refused new implants, while the prosthesis remained in function supported by three implants in each patient. In two patients, two prostheses failed (one prosthesis in each patient) following implant failures, rendering a prosthetic survival rate of 98.4% (dehiscence group: 98.9%; fenestration group: 96.4%).

Considering the patient as the unit of analysis, the overall implant cumulative survival at 10 years was 94.1%, with 95.6% and 88.1% for the dehiscence and fenestration groups, respectively (Kaplan–Meier, Table 3). Implant failure was higher in smokers, with an implant cumulative survival of 87.7% for smokers with dehiscence (compared to 98.6% for nonsmokers) and 74.6% for smokers with fenestrations (compared to 95.7% for nonsmokers) (Figure 4), as well as in patients with systemic conditions (93.3% in the dehiscence group compared to 100% in healthy patients; 86.7% in the fenestration group compared to 91.7% in healthy patients).

Considering the implant as the unit of analysis, the overall cumulative survival rate at 10 years was 96.2%, with 97.2% for the dehiscence group and 90.0% for the fenestration group (Table 4). The overall implant cumulative success rate was 93.5%, with 94.6% and 87.6% for the dehiscence and fenestration groups, respectively (Table 5).

### 3.3. Marginal Bone Loss

Overall, the percentages of available radiographs for performing the marginal bone resorption measurements were 78% at 5 years and 90% at 10 years. The average (standard deviation) marginal bone loss at 5 years was 1.22 mm (0.66 mm), with 1.25 mm (0.69 mm) for the dehiscence group and 1.22 mm (0.66 mm) for the fenestration group (Figure 5).

The average (standard deviation) marginal bone loss at 10 years was 1.53 mm (0.75 mm), with 1.49 mm (1.34 mm) for the dehiscence group and 1.53 mm (0.75 mm) for the fenestration group (Figure 6).

### 3.4. Complications

Biological complications occurred for a total of 18 implants (9.4%) in 18 patients (14%): 13 implants (9.1%) in the dehiscence group and 6 implants (12.5%) in the fenestration group (one implant had both dehiscence and fenestration). The complications included suppuration (2 implants), infection (2 implants), fistula (1 implant), and peri-implant pathology (13 implants). The two incidences of suppuration and the incidence of fistula were resolved by utilizing non-surgical therapy (scaling with an ultrasonic scaler device, irrigating with 0.2% chlorhexidine gel, and prescribing the same solution for the patient in daily care, followed by a re-evaluation) and antibiotic administration. One infection was resolved non-surgically, while the other incidence was not resolved (implant considered unsuccessful). The incidences of peri-implant pathology were resolved in five implants through non-surgical intervention and surgically in one implant through an open flap, the removal of granulation tissue, and a decontamination of the implant surface both mechanically using an ultrasonic scaler and chemically through a polish using a brush attached to the contrangle and 0.2% chlorhexidine and saline irrigation; the mucosal flap, apically positioned, was sutured using 4/0 non-resorbable sutures (Braun Silkam4-0, Aesculap, Tuttlinged, Germany), and systemic antibiotics were prescribed to the patient (amoxicillin 875 mg + clavulanic acid 125 mg, Labesfal, Campo de Besteiros, Portugal). In seven implants, peri-implant pathology was not resolved non-surgically or surgically, with two of the implants being removed (considered failures) and five implants remaining in function (considered unsuccessful). Seven of the unresolved biological complications (87.5%) were implants with dehiscence (one implant with both dehiscence and fenestration).

## 4. Discussion

The present study registered a successful outcome of full-arch restorations with the All-on-4 concept for implants inserted with dehiscence or fenestration after 10 years of follow-up. Previous long-term publications using the same rehabilitation method registered high cumulative implant survival and success rates at the same point of evaluation (10 years). In the maxilla, a study evaluating the outcome between 5 and 13 years of 1072 patients with 4288 implants registered 96.4% and 95.6% of cumulative implant survival and success rates, respectively, together with an average marginal bone loss of 1.7 mm [6]; for the mandible, the evaluation between 10 and 18 years of 471 patients with 1884 implants registered a cumulative survival rate of 96.9%, a cumulative success rate of 95.9%, and an average marginal bone loss of 1.67 mm [7]. The results of the present study imply that full-arch rehabilitations can also be performed with predictability in extreme situations rather than being applied only in the presence of generally accepted inclusion criteria for implant rehabilitation. Nevertheless, despite a favorable marginal bone loss outcome at 5 and 10 years, the overall implant success outcome of the present study was lower when compared to those publications [6,7], which is a result significantly influenced by the fenestration group, with markedly increased failure (10%) and unsuccessful (above 12%) implant outcomes. The occurrence of dehiscence or fenestration is considered an accident [25] or complication [21] of adequate treatment, excluded from the listings of failures and independent from surgical precautions or skills [21]. Usually, the presence of these conditions is attributed to bone-plate thickness deficiency [26], illustrative of challenging conditions for full-arch implant-supported rehabilitations, which may partly explain the lower success outcome. As for the fenestration subgroup, these results concur with previous case reports [27] with a higher incidence of short-term implant losses in these situations. In both conditions, the results follow previously reported pattens for the short-term outcome, with a high survival rate for implants placed with dehiscence, as registered in a clinical trial [15], and a higher incidence of implant failures for implants with fenestration (the majority up to ~2 years postsurgery), as observed in case reports [27]. The type of complementary therapeutic approach (bone augmentation at implants with dehiscence and fenestrations, with or without the use of membranes) does not seem to significantly impact survival outcomes [20], which is a result that is in line with the present study, with high survival rates for implants with dehiscence and no complementary therapeutical approaches, while the outcome of implants with fenestrations was independent of the use of autogenous bone graft.

Smoking was previously registered as a risk indicator for implant failure in a systematic review and meta-analysis, with a 2.92-fold increased odds ratio compared to nonsmokers [28]. Previous publications on the same treatment modality (All-on-4) also registered a significant negative influence of smoking habits with increased implant failure, excess marginal bone loss, or biological complications [6,7]. A clinical trial investigating the outcome of implants with dehiscence placed with simultaneous guided bone regeneration between 22 and 24 years of follow-up registered a significant negative effect of smoking on survival [29]. In the present study, smoking impaired implant survival in both groups with a ~11% decrease for implants with dehiscence and 21% decrease for implants with fenestrations, with a higher percentage of smokers present in the fenestration group where an increased failure rate was registered. Moreover, most smokers in both groups registered late implant failures beyond the osseointegration phase (all three smokers from the dehiscence group and two in three smokers from the fenestration group). It can be hypothesized that the difference in survival could be at least partially due to a confounding effect of smoking rather than the effect of dehiscence or fenestration. Confounding variables are those that may compete with the exposure of interest (in this case, dehiscence and fenestration) in explaining the outcome of the study (implant survival) [30]. To be considered a confounding variable, the variable needs to meet two conditions: it must be correlated with the independent variable (the majority of patients from the fenestration group were smokers) and causally related to the dependent variable (smoking related with implant failure) [28]. Nevertheless, due to a lack of statistical power, it was not possible to assess this evaluation in the present study; therefore, this should be further investigated in studies using statistical models to control for confounding with an adequate sample size.

The biological complication rate was reduced but concentrated on the dehiscence subgroup (mainly peri-implant pathology), particularly the unresolved complications. The presence of dehiscence implies a potential settlement of the bone level around the implant’s middle third at baseline, a condition previously registered as a risk indicator for peri-implant pathology in both inferential [31] and multivariable analysis [23]. A previous study evaluating the impact and accuracy of a risk algorithm for the incidence of peri-implant pathology registered that patients with implants whose bone level was at the middle third have a 14-fold increase in the probability of disease, estimating that the prevention of this condition could potentially suppress 31% of the cases of peri-implant pathology based on the attributable fraction [23]. Furthermore, a clinical trial [15] evaluating the outcome of implants with buccal dehiscence registered a significant increase in clinical attachment loss and probing pocket depths for these implants when compared to implants inserted in healed sites. The implementation of a maintenance protocol, with programmed recall appointments for clinical diagnosis, prophylaxis, oral hygiene instructions, and management of the implant–abutment connection through disinfection [32], is considered paramount for good long-term outcomes. In the present study, the implementation of a maintenance protocol and recall appointments allowed performing close monitoring of the patients that otherwise could have potentially resulted in a further increased implant failure rate.

The dropout rate of 20% was small considering the 10-year span of this clinical study and accounted for good internal validity. Nevertheless, this study had limitations that warrant discussion, including the retrospective evaluation; the convenience sample; the lack of a control group without bone defects; the small sample size that disabled inferential analysis; the lack of stratification for smoking habits; not collecting data on mechanical complications; and being performed in a single center and with one implant system, which imply caution in the extrapolation of the results. Furthermore, the significant difference in age between the sample lost to follow-up and the fully analyzed sample might present a potential bias in the underestimation of the implant and prosthetic survival outcomes, warranting caution in the interpretation of the results.

Future studies with larger samples and multivariable analysis should be performed to control potential confounding effects when evaluating the outcome of these rehabilitations and to allow subsample analytical methods and age and gender discrimination.

## 5. Conclusions

Considering the limitations of the present study, the long-term outcome of implants inserted with dehiscence and fenestration for full-arch rehabilitations with the All-on-4^®^ concept is possible with overall high survival rates and low average marginal bone resorption; nonetheless, implants with fenestrations yielded lower survival and success rates. Smoking exerted a significant negative effect on implant survival. Implants with dehiscence suffered most from biological complications that could not be resolved.

## Figures and Tables

**Figure 1 jcm-11-01939-f001:**
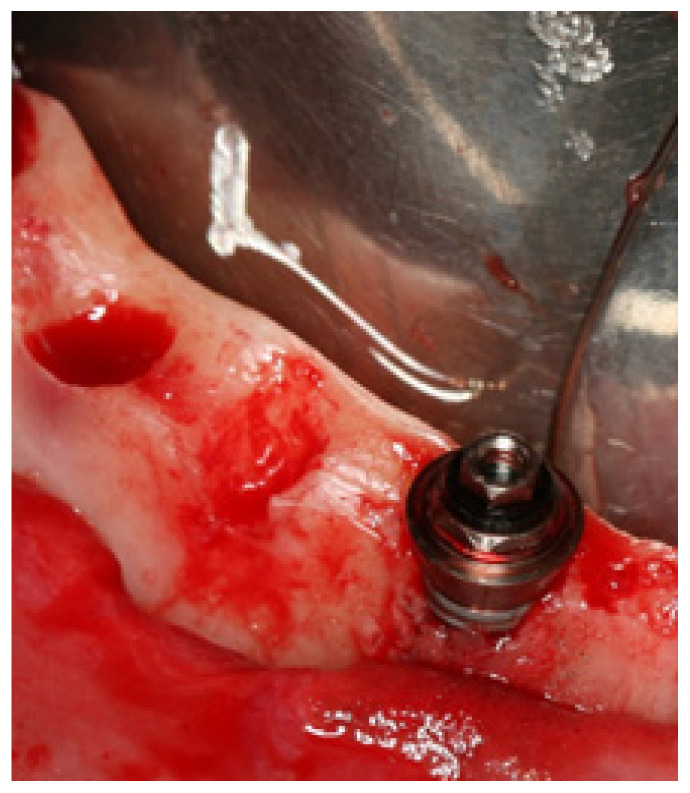
Intraoral photograph illustrating dehiscence: the absence of alveolar bone on the buccal or palatine/lingual aspects, leaving an oval-shaped defect and implant exposure apically from the implant–abutment junction.

**Figure 2 jcm-11-01939-f002:**
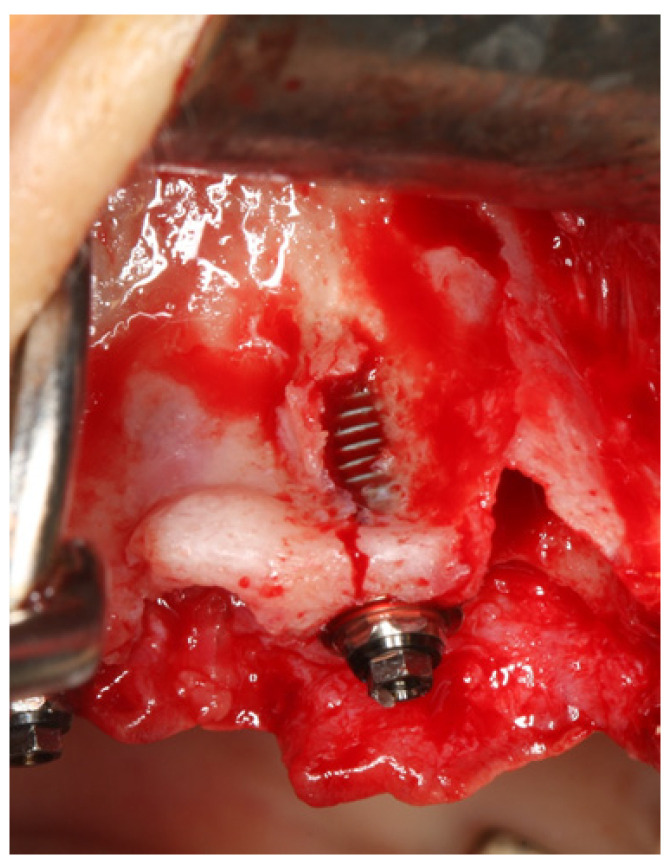
Intraoral photograph illustrating a fenestration, defined as a buccal or lingual window defect leaving marginal bone in situ.

**Figure 3 jcm-11-01939-f003:**
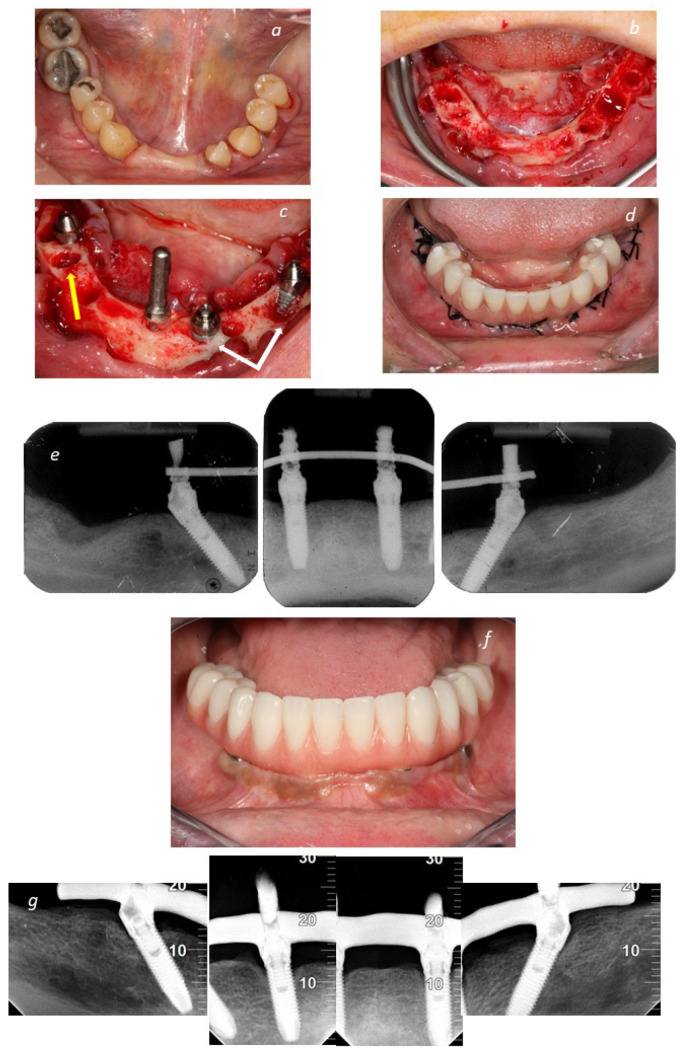
Clinical and radiographical displays illustrations of patients included in the study: (**a**) pretreatment intraoral occlusal view of the mandible; (**b**) perioperative occlusal view following tooth extraction; (**c**) perioperative clinical photograph illustrating the implants. Note the dehiscence in the two implants of the third quadrant (anterior axial implant and posterior tilted implant on the right pointed by the white arrows) and the fenestration on the implant of the fourth quadrant (posterior tilted implant on the left pointed by the yellow arrow); (**d**) prosthesis connected on the same day of surgery achieving immediate function; (**e**) baseline periapical radiographs of the same patient; (**f)** clinical photograph illustrating the 10 year follow-up of the same patient; (**g**) 10-year periapical radiographs of the same patient.

**Figure 4 jcm-11-01939-f004:**
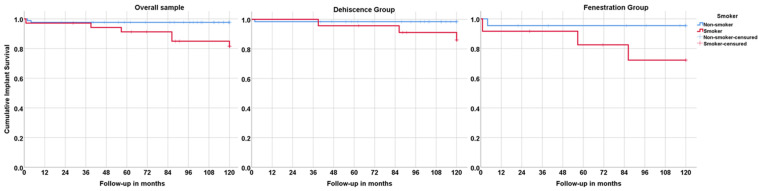
Graphical illustration of the cumulative implant survival estimation using the patient as the unit of analysis. Note the significant negative influence of smoking habits on implant survival overall and in both groups.

**Figure 5 jcm-11-01939-f005:**
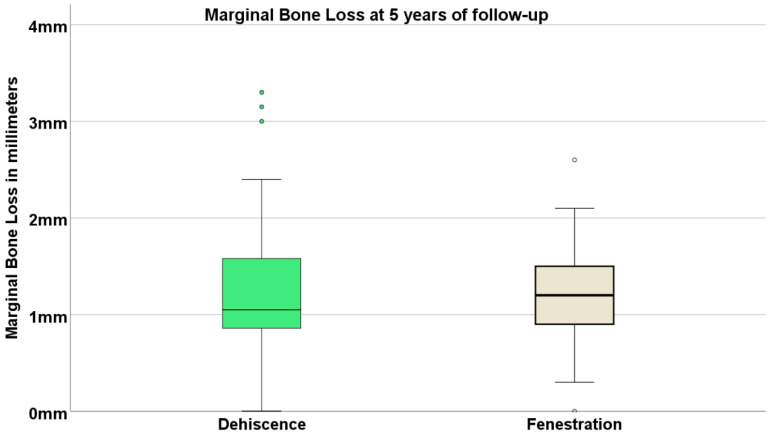
Boxplot illustrating marginal bone loss at 5 years in both groups. The lower box edge represents 25% of the sample; the upper box edge represents 75% of the sample; the horizontal black line represents the median; dots represent outlier values. Note that the median in both groups was localized below 1.25 mm at 5 years.

**Figure 6 jcm-11-01939-f006:**
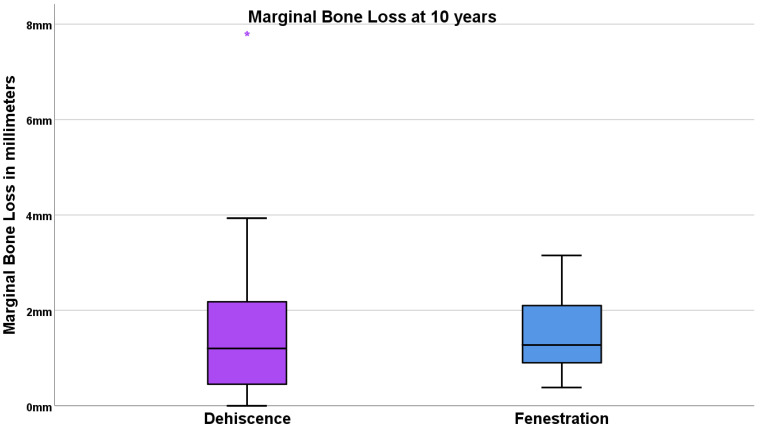
Boxplot illustrating the marginal bone loss at 10 years in both groups. The lower box edge represents 25% of the sample; the upper box edge represents 75% of the sample; the horizontal black line represents the median; star represents an outlier value. Note that the median in both groups was localized below 1.3 mm at 10 years.

**Table 1 jcm-11-01939-t001:** Overall medical status distribution according to the International Classification of Disease, version 11 (ICD-11) and implant distribution in the sample.

ICD-11 Classification	ICD-11 Group Description		Examples	*n* Patients
1	Certain infectious or parasitic diseases		(HIV, hepatitis)	2
2	Neoplasms		(Cancer)	2
4	Diseases of the immune system		(Lupus)	1
5	Endocrine, nutritional, or metabolic diseases		(Diabetes, Hyperthyroidism)	12
6	Mental, behavioral, or neurodevelopmental disorders		(Depressive disorder)	3
8	Diseases of the nervous system		(Multiple sclerosis, epilepsy)	2
11	Diseases of the circulatory system		(Hypertension, angina)	31
12	Diseases of the respiratory system		(Asthma)	1
13	Diseases of the digestive system		(Heavy bruxer)	3
15	Diseases of the musculoskeletal system or connective tissue		(Osteoporosis)	6
16	Diseases of the genitourinary system		(Prostate disorder)	1
18	Pregnancy, childbirth or the puerperium		(Hysterectomy)	2
24	Factors influencing health status		(Smoking)	42
26	Nasal sinusitis disorder		(Sinusitis)	1
	Healthy		—	44
Totals	* 25 patients presented with more than a single condition in a total of 79 patients.			123 *
**Implant Distribution According to Platform and Length**				
**Type**	**Diameter**	**Length**	***n* Implants (*n* Lost; Position; Follow-Up; Group)**	
Mk III	3.75	15 mm	1 (1; #42; 2 months; dehiscence)	
	4	15 mm	9	
Mk IV	4	15 mm	4 (1; #22; 2 months; fenestration)	
NobelSpeedy Replace	3.5 mm	15 mm	1	
NobelSpeedy Groovy	3.3	10 mm	1	
		11.5 mm	4	
		13 mm	1	
		15 mm	1	
	4	8.5 mm	5 (1; #12; 73 months; dehiscence)	
		10 mm	3	
		11.5 mm	14	
		13 mm	23 (1; #44; 86 months; dehiscence-fenestration)	
		15 mm	72 (1; #44; 4 months; fenestration)	
		18 mm	53 (2; (#15; 39 months; dehiscence) (#25; 57 months; fenestration))	
Total			192 (7)	

**Table 2 jcm-11-01939-t002:** Distribution of patients and implants according to the implant’s defect localization.

	Total	Dehiscence			Fenestration		
		Total	Vestibular	Lingual/Palatal	Total	Vestibular	Lingual/Palatal
Number of patients ^1^	109	95	46	53	36	28	9
Number of patients (smokers) ^2^	35	25	14	12	13	9	4
Number of implants ^3^ (lost)	192	152	71 (4 ^4^)	83	42	35 (3)	7 (2 ^4^)

^1^ Eight patients with dehiscence and fenestration and multiple implants with dehiscence on several sites; ^2^ three patients with dehiscence and fenestration; ^3^ two implants with a dehiscence and fenestration; ^4^ one implant with dehiscence in vestibular and fenestration in palatal region.

**Table 3 jcm-11-01939-t003:** Cumulative implant survival rate using the patient as the unit of analysis: total sample and per group estimations (Kaplan–Meier product limit estimator).

Time (Months)	Status (0 = Survival; 1 = Failure)	Cumulative Proportion Surviving at the Time		*N* of Cumulative Events	*N* of Patients at Risk
		Estimate	Std. Error		
Total Sample					
0	0	.	.	0	123
1	1	0.992	0.008	1	122
2	1	0.984	0.011	2	121
4	1	0.976	0.014	3	120
24	0	.	.	3	119
29	0	.	.	3	118
39	1	0.967	0.016	4	117
40	0	.	.	4	115
48	0	.	.	4	113
55	0	.	.	4	112
57	1	0.959	0.018	5	111
60	0	.	.	5	110
72	0	.	.	5	106
73	1	0.950	0.020	6	105
86	1	0.941	0.022	7	104
96	0	.	.	7	100
108	0	.	.	7	95
120	0	.	.	7	90
Dehiscence Group					
0	0	.	.	0	95
2	1	0.989	0.010	1	94
39	1	0.979	0.015	2	93
48	0	.	.	2	90
60	0	.	.	2	88
72	0	.	.	2	85
73	1	0.967	0.019	3	84
86	1	0.956	0.022	4	83
96	0	.	.	4	79
108	0	.	.	4	75
111	0	.	.	4	73
117	0	.	.	4	71
120	0	.	.	4	70
Fenestration Group					
0	0	.	.	0	36
1	1	0.972	0.027	1	35
4	1	0.944	0.038	2	34
24	0	.	.	2	33
36	0	.	.	2	32
48	0	.	.	2	31
56	1	0.914	0.048	3	30
72	0	.	.	3	29
85	0	.	.	3	28
86	1	0.881	0.056	4	27
117	0	.	.	4	25
120	0	.	.	4	24

**Table 4 jcm-11-01939-t004:** Life tables for cumulative implant survival rate using the implant as the unit of analysis: dehiscence and fenestration groups.

**Dehiscence Group**					
**Time (Months)**	**Number of Implants**	**Number Lost to Follow-Up**	**Number of Failures**	**Survival Rate**	**Cumulative Survival Rate**
Baseline	152	0	1	99.3%	99.3%
1 year	151	0	0	100%	99.3%
2 years	151	0	0	100%	99.3%
3 years	151	5	1	99.3%	98.7%
4 years	145	4	0	100%	98.7%
5 years	141	3	0	100%	98.7%
6 years	138	0	1	99.3%	98.0%
7 years	137	6	1	99.3%	97.2%
8 years	130	5	0	100%	97.2%
9 years	125	6	0	100%	97.2%
10 years	119	0	0	100%	97.2%
**Fenestration Group**					
**Time (Months)**	**Number of Implants**	**Number Lost to Follow-Up**	**Number of Failures**	**Survival Rate**	**Cumulative Survival Rate**
Baseline	42	0	2	95.2%	95.2%
1 year	40	1	0	100%	95.2%
2 years	39	1	0	100%	95.2%
3 years	38	1	0	100%	95.2%
4 years	37	0	1	97.3%	92.7%
5 years	36	1	0	100%	92.7%
6 years	35	0	0	100%	92.7%
7 years	35	1	1	97.1%	90.0%
8 years	33	1	0	100%	90.0%
9 years	32	2	0	100%	90.0%
10 years	30	0	0	100%	90.0%

**Table 5 jcm-11-01939-t005:** Life tables for cumulative implant success rate using the implant as the unit of analysis: dehiscence and fenestration groups.

**Dehiscence Group**					
**Time (Months)**	**Number of Implants**	**Number Lost to Follow-Up**	**Number of Failures**	**Survival Rate**	**Cumulative Survival Rate**
Baseline	152	0	1	99.3%	99.3%
1 year	151	0	3	98.0%	97.4%
2 years	148	0	1	99.3%	96.7%
3 years	147	5	0	100%	96.7%
4 years	142	4	1	99.3%	96.0%
5 years	137	3	0	100%	96.0%
6 years	134	0	1	99.3%	95.3%
7 years	133	6	1	99.2%	94.6%
8 years	126	5	0	100%	94.6%
9 years	121	5	0	100%	94.6%
10 years	116	0	0	100%	94.6%
**Fenestration Group**					
**Time (Months)**	**Number of Implants**	**Number Lost to Follow-Up**	**Number of Failures**	**Survival Rate**	**Cumulative Survival Rate**
Baseline	42	0	2	95.2%	95.2%
1 year	40	1	0	100%	95.2%
2 years	39	1	1	97.4%	92.8%
3 years	37	1	0	100%	92.8%
4 years	36	0	1	97.2%	90.2%
5 years	35	0	0	100%	90.2%
6 years	35	1	1	97.1%	87.6%
7 years	33	1	0	100%	87.6%
8 years	32	1	0	100%	87.6%
9 years	31	1	0	100%	87.6%
10 years	30	0	0	100%	87.6%

## Data Availability

Data are available from a public repository, https://doi.org/10.17605/OSF.IO/E3ZRC, accessed on 17 March 2022.

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
