# Peer review of "The 10 Year Outcomes of Implants Inserted with Dehiscence or Fenestrations in the Rehabilitation of Completely Edentulous Jaws with the All-on-4 Concept"

_jcm, 2022, doi:10.3390/jcm11071939_

Round 1

Reviewer 1 Report

The manuscript is relevant for the field and presented in a well-structured manner.

Although, the manuscript seems to be scientifically sound. However, the experimental design created a doubt. As per the authors, this study is retrospective cohort study and the ethical clearance was obtained in 2017. So after 2017, the follow up period is 5 years only. But in the study, you have shown the follow up of 10 years. Ethically the baseline data should be after ethical clearance only. Please elaborate in details the design of study. For example, suppose in one patient, the implants were placed in 2003, then their baseline IOPA was taken when? Its very confusing. Please explain.

Although all the figures/tables/images/schemes appropriate but for clarity of the reader do provide some illustrative diagram of fenestration and dehiscence if possible.

Other sections are written well

Reviewer 2 Report

Dear Editor,

Regarding the submitted manuscript “Ten years outcome of implants inserted with dehiscences or fenestrations in the rehabilitation of complete edentulous jaws through the All-on-4 concept.” this review will be divided in overall and detailed appreciation.  The presented study is intended to be a retrospective study.

Overall appreciation

The manuscript reflects a study which intends to evaluate the influence of dehiscences or fenestrations in full arch rehabilitations. Although the authors addressed an important subject there are some concerns that I think the authors should address before it is considered for publication

  • English – I advise a revision by a native tongue speaker since I found some grammatical mistakes in the abstract and introduction
  • Study design characteristics? –Sample recruitment (convenience? how the authors ensured that all fenestrations and dehiscence’s were identified) , Sample size determination? Power analysis?
  • Was the sample frame appropriate to address the target population?
    1. I think that some details need better clarification. For example, the authors mention heavy bruxers, how was that variable identified? The presence of parafunctional habits (such as bruxism) detected prior to initiation of the final rehabilitation sequence could present as an important confounding variable since these patients are known to present with a higher risk for mechanical complications. (Mikeli and Walter, 2016) and authors need to address method used for detection.
    2. The authors do not report smoking stratification being that several articles report number of cigarettes influence.
    3. Authors need to mention acrylic teeth used in the provisionals since they only mention the pink acrylic.
    4. What were the professional oral hygiene habits/controls for the patients ? Were the screened 1??3? times per year? How many did not comply with the set timetables. Literature reports the professional controls influence in implant success and it should be addressed.
  • Were the samples measured more than once in different times and the intra and correlation calculated (ICC)? Without these data we cannot know if the results are dependent on the observer.
  • Was the sample size adequate? – I cannot find any sample size calculation for the analyzed modifiers or a power analysis. I believe that it needs to be reported since the sample size presented is rather small.
  • According to the Kaplan-Meier curves in the Figure 2, I think the proportionality assumption for K-M analysis or Cox proportional model is violated.  The authors need to address this point.
  • Was the condition measured in a standard, reliable way for all participants? - How many evaluators determined the lengths? Were they calibrated? Were the samples measured more that once in different times and the intra and interclass correlation calculated (ICC)? Without these data we cannot know if the results are dependent on the observers

Reviewer 3 Report

Dear authors, full arch rehabilitations remain one of the most interesting challenges for clinicians and researchers. To date, the world literature is very attentive to the subject, demonstrating how many proposed protocols bring excellent long-term results. However, ideal clinical situations are not always achievable. For this I congratulate the authors for the selected topic. The job is well done, however it needs some changes.

The introduction is well written. It presents the topic and the gap present in the literature. The hypothesis of the study could be added after the aim of the study.

 In materials and methods the inclusion and exclusion criteria should be expressed more broadly. Were heavy smokers or patients with relevant systemic diseases included? Have minimal amounts of residual bone and soft tissue been considered? This could change the study results.

Figure 1 I recommend adding a clinical photo to the follow-up (10Y)

Information on patient follow ups could be added. Have the patients been followed up over time? Have specific maintenance protocols been implemented? Could the periodic monitoring of patients have influenced the outcome of the study? Have the prosthetic complications been recorded?

The results are clear. I recommend inserting a summary table given the vast amount of data collected. It would be more intuitive for the reader. By inserting total implants, different implants with bone defects, number of smoking patients, palatal or vestibular defect… etc 

It would be interesting to insert further clinical images of implants with dehiscences and fenestrations. And try to classify patients and their defects to better differenziate the sample. doi: 10.1186 / s12880-021-00557-9.

At the same time ask the authors why implants without bone defects were excluded from the evaluation. It could be interesting To have a control group on the same sample and related complications and 10-year survival.

Furthermore, dividing the vestibular from the palatal dehiscences in the analysis of the results could add value to the study.

The discussion is well done. I stress again the need to include a paragraph on disinfection and maintenance protocols, on the periodic recall of patients and on the management of the connection during the different time points doi: 10.1155 / 2018/5326340. These factors among others can influence long-term survival and should be taken into consideration

Round 2

Reviewer 2 Report

The authors adressed the  remarks and it is this reviewer opinion.

Author Response

The authors adressed the  remarks and it is this reviewer opinion.

Response: The authors thank the Reviewer for all the work put into our manuscript.